# The Therapeutic Effect and the Potential Mechanism of Flavonoids and Phenolics of *Moringa oleifera* Lam. Leaves against Hyperuricemia Mice

**DOI:** 10.3390/molecules27238237

**Published:** 2022-11-25

**Authors:** Xiaowei Luo, Lipeng Zhou, Shukai Wang, Jing Yuan, Zihao Chang, Qian Hu, Yinxin Chen, Yuqi Liu, Ya Huang, Baojin Wang, Ye Gao, Zhaohui Wang, Yitong Cui, Yue Liu, Lanzhen Zhang

**Affiliations:** School of Chinese Pharmacy, Beijing University of Chinese Medicine, Beijing 102488, China

**Keywords:** *Moringa oleifera* Lam. leaves, flavonoids, phenolics, metabolomics, molecular docking, hyperuricemia

## Abstract

The aim of this study is to evaluate the anti-hyperuricemia effect and clarify the possible mechanisms of flavonoids and phenolics of MOL (MOL-FP) in mice. Hyperuricemia mice were generated via intraperitoneal (i.p.) administration of potassium oxonate (PO) and oral gavage (p.o.) of hypoxanthine (HX). Serum uric acid (UA), weight, serum XO activity, hepatic XO activity, urea nitrogen (BUN), creatinine (CRE), serum AST level, serum ALT level, mRNA expression of renal urate-anion transporter 1 (URAT1), glucose transporter 9 (GLUT9), organic anion transporters 1 (OAT1), organic anion transporters 3 (OAT3), and ATP-binding cassette transporter G2 (ABCG2) were determined. The molecular docking was conducted using AutoDock Vina 1.2.0 to screen potential XO inhibitors in MOL-FP. Serum metabolomics was established to collect the metabolic profiles of mice and explore the metabolic changes that occurred after MOL-FP treatment. MOL-FP could notably reduce the serum UA level of hyperuricemia mice by inhibiting XO activity and regulating renal urate transporters. Molecular docking studies indicated that 5-*p*-coumaroylquinic acid, 3-*p*-coumaroylquinic acid, and catechin could be potential XO inhibitors. Besides, MOL-FP prevented the pathological process of hyperuricemia by regulating biomarkers associated with purine metabolism, amino acid metabolism, and lipid metabolism.

## 1. Introduction

Hyperuricemia, as the serum UA concentration exceeds the normal limit (≥420 μmol/L for males and ≥360 μmol/L for females), is a common disease with a worldwide distribution and continues to be a health problem. The global morbidity of hyperuricemia in the population ranges from 8.4% to 37.2 [1,2,3]. The in-hospital mortality was 6.2% (69 out of 1117), and the patients with hyperuricemia carried a higher risk of in-hospital death [4]. Clinical studies have shown that the risk of gout increases with the degree and duration of hyperuricemia. The serum urate levels at concentrations above the solubility point of uric acid may result in the formation of uric acid crystals in the joints, causing gouty arthritis [5]. Apart from that, uric acid is related to an increased risk of cardiovascular disorder, nephrolithiasis, and diabetes [6].

The increased incidence of hyperuricemia is related to excessive production and limited excretion of uric acid. Xanthine oxidase (XO), which catalyzes the conversion of hypoxanthine to xanthine and xanthine to uric acid, is a key enzyme in the purine catabolic pathway [7]. Therefore, it is an important treatment in urate-lowering therapy through the suppression of xanthine oxidase activity [8]. Allopurinol, a xanthine oxidase inhibitor, is an effective urate-lowering drug and has been employed for the treatment of hyperuricemia for over 50 years [9]. However, allopurinol is a frequent cause of severe cutaneous adverse reactions (SCAR), which include hypersensitivity syndrome, Stevens–Johnson syndrome, toxic epidermal necrolysis, eosinophilia, and systemic rash manifestations [10].

In addition, hyperuricemia, is more common, caused by the reduced renal excretion of urate [11]. The urate filtered by renal glomeruli is reabsorbed by transepithelial transport in the renal proximal tubule, with around 10% fractional excretion. This means reabsorption largely dominates secretion in the kidney. The urate anion transporter 1 (URAT1) and glucose transporter 9 (GLUT9) play a crucial role in urate reabsorption [7]. Benzbromarone and probenecid, as uricosuric drugs, was a second-line agent for the treatment of hyperuricemia. These medications down-regulate urate transporters URAT1 or GLUT9 expression, promoting uric acid excretion [12]. Unfortunately, the uricosuric drugs, in patients with Hepato-renal dysfunction, may cause severe liver or kidney injury [13]. Thus, there is highly warranted for novel anti-hyperuricemia agents with the absence of toxic or side effects.

*Moringa oleifera* Lam (MO), which belongs to the Moringaceae family, is cultivated in many tropical and subtropical regions of the world [14]. MO has various medicinal uses including anti-inflammatory, antioxidant, anti-cancer, hepatoprotective, neuroprotective, hypoglycemic, and blood lipid-reducing functions, which have long been recognized in the Ayurvedic and Unani systems of medicine [15]. The multiple parts of MO are valuable for medicinal or food ingredients, and *Moringa oleifera* Lam. leaf (MOL) is one of the dominant utilized parts of the plant. MOL was found to contain a huge quantity of phenolic compounds, principally flavonoids, phenolics, and their glycosides [16]. MOL has been used as a good source of flavonoids and phenolics [17]. According to previous research, xanthine oxidase (XO) can be blocked by phenolics, thereby reducing the synthesis of UA, and combating HUA disorders. In addition, phenolics can inhibit the renal reabsorption of urate and improve the secretion of UA [18]. In previous studies, the phenolic and peptide fractions of MOL ameliorated hyperuricemia and metabolic disorders [19]. In the current study, we have studied the potential mechanism between MOL-FP, XO, and the regulation of endogenous metabolic components in hyperuricemia, which is rarely mentioned.

The purpose of this study was to elucidate the potential mechanism of the flavonoid and phenolics from MOL-FP in the treatment of hyperuricemia by molecular docking and metabolomics. Firstly, the crude extract of MOL was purified to yield a flavonoid and phenolics fraction. The hyperuricemia mouse model was induced by potassium oxonate intraperitoneal injection and hypoxanthine intragastric administration. It was analyzed that the multiple targets of MOL-FP to low uric acid by identifying the expression of XO and renal transporters. The binding mechanism of small molecule compounds and XO was explained by molecular docking. The effects on hepatic and renal function were evaluated by biochemical assay and histopathology of tissues. Furthermore, serum metabolomics was employed using UPLC-MS to explore the potential biomarkers and determine the overall metabolic changes of MOL-FP in hyperuricemia mice. PCA was applied to distinguish endogenous metabolites of normal, hyperuricemia, and MOL-FP-treated mice. OPLS-DA was related to finding potential biomarkers. Metabolic pathway analysis was conducted to elucidate the metabolic network regulated by MOL-FP. This study will support a better understanding of the difference in potential mechanisms of MOL-FP in improving hyperuricemia and provide alternative strategies for the prevention and treatment of hyperuricemia.

## 2. Results

### 2.1. Effects of MOL-FP on the Levels of UA, Weight, and the Activity of XO in Hyperuricemia Mice

MOL-FP might ameliorate hyperuricemia via multiple targets. The inhibition of XO, promotion of urate excretion, and improvement of renal dysfunction are potent strategies for regulating hyperuricemia and metabolic disorders. The serum UA levels of the Model group were significantly higher than that of the control group after PO and HX treatment for 15 consecutive days (Figure 1A), demonstrating the successful induction of a mouse hyperuricemia model. The oral administration of AP (10 mg/kg) and MOL-FP (600 mg, 1200 mg, and 2400 mg/kg) could significantly reduce the serum UA levels of hyperuricemia mice compared with the Model group (Figure 1A), demonstrating the significant anti-hyperuricemia effect of MOL-FP.

The body weights of all the mice increased steadily over 15 days and hyperuricemia induced by PO and HX could slightly decrease the body weights of mice (Figure 1B). However, AP and MOL-FP treatment could not alleviate the symptoms. Mice in the AP group had disturbed body hair and were in a bad mental state during the experimental period.

The inhibition of XO activity facilitates the reduction of UA levels. After PO and HX treatment, serum XO activity noticeably increased and significantly decreased after AP treatment. These results were consistent with the previous study [20]. MOL-FP treatment significantly decreased the XO activities of hyperuricemia mice (Figure 1C), which indicated that MOL-FP played an important role in inhibiting serum XO activity for the regulation of hyperuricemia. XO functional activity in the liver was also increased in hyperuricemia mice and significantly decreased in MOL-FP administration mice (Figure 1D), indicating that MOL-FP administration was able to antagonize hepatic XO activity in hyperuricemia mice.

### 2.2. Effects of MOL-FP on the Related Transporters in Renal Tissue

In the kidney, urate excretion is controlled by a group of urate transporters in the proximal tubule [7]. Depending on the site of action of the kidney, urate transport proteins are divided into two main categories: reabsorbed proteins represented by GLUT9 and URAT1, and secreted proteins represented by OAT1, OAT3, and ABCG2 [11]. In this study, the expression of the urate transporter was detected by RT-qPCR to evaluate urate excretion in renal tissue. PO and HX significantly upregulated the expression of renal URAT1 and GLUT9 and downregulated the expression of renal ABCG2, OAT1, and OAT3, suggesting dysfunctional UA excretion in the kidney. MOL-FP significantly up-regulated the expression of ABCG2, OAT1, and OAT3 and down-regulated the expression of URAT1 and GLUT9 (Figure 2). The results indicated that MOL-FP accelerated the excretion of UA and effectively reduced the serum UA level in hyperuricemia mice by up-regulating the expression of ABCG2 and down-regulating the expression of URAT1 and GLUT9.

### 2.3. Effects of MOL-FP on Liver and Kidney Injury in Mice with Hyperuricemia

The high level of serum UA is associated with liver and kidney injury. Compared with the Control group, the mice of the Model group showed several visible histological changes, including proximal tubule necrosis and dilation, renal tubule swelling, cytoplasmic vacuolization, and inconspicuous boundaries between adjacent proximal tubule cells (Figure 3A), suggesting the renal damage caused by PO and HX. BUN and CRE are laboratory signs of kidney injury. After PO and HX administration, the serum BUN levels were remarkably increased compared with the Control group. MOL-FP (MD and HD) significantly reduced BUN levels (Figure 1G). In addition, the serum CRE levels of mice slightly increased after PO treatment and decreased significantly after AP and MOL-FP treatment (Figure 1H). H&E staining showed that the structure of hepatic lobules was intact, and the hepatic cords were arranged radially and regularly around the central vein without obvious lesions in the Control group. After PO and HX treatment, the arrangement of hepatic cords was disordered, and the liver cytoplasm was vacuolated, accompanied by inflammatory cell infiltration in the Model group. the HD group found significant improvement in liver tissue (Figure 3B). ALT and AST (markers of liver injury) are specific aminotransferases that are expressed in the liver [21]. Serum AST and ALT levels of the Model group notably increased compared with the Control group, indicating liver damage in hyperuricemia mice. MOL-FP (HD) treatment significantly decreased ALT and AST levels in serum, suggesting a possible hepatoprotective effect (Figure 1E,F). These results showed that the MOL-FP treatment group could reverse the liver injury of PO and HX-induced hyperuricemia mice to some extent.

### 2.4. Potential Inhibitors of XO in MOL-FP

To identify potential compounds in MOL-FP that contribute to hyperuricemia, XO (PDBID:3NVW) was selected as the target to screen the chemical component of MOL-FP densified by us. Compared with Guanine (the binding energy was −6.7 kcal/mol), most of the compounds had lower energy and better affinity with XO in different poses (Appendix A). The three top-rank compounds all occupied the active tunnel for substrate entering XO, which may inhibit the substrate from entering (Figure 4). Of the three, 5-*p*-coumaroylquinic acid had the lowest binding energy of −9.9 kcal/mol. It formed two hydrogen bonds with THR1010 and VAL1011 interacting with the oxygen atoms of a hydroxyl attached to the aromatic ring, and GLN767 formed hydrogen bonding with the oxygen atoms of a hydroxyl attached to the quinic ring. Moreover, it bonded with PHE914 and PHE1009 through Pi–Pi Stacked interactions, and ALA1079 and ALA1078 through Pi-Alkyl interactions. 3-*p*-coumaroylquinic acid was ranked second, with a binding energy of −8.1 kcal/mol. ARG880 and SER876 were involved in the H bond interactions. ARG880 forms H bonds with oxygen atoms belonging to the hydroxyl group on the aromatic ring of the compound. SER876 forms H bonds to oxygen atoms owned by the carbonyl group of the compound. The binding energy of catechin was −7.8 kcal/mol. In the interaction pattern, the residues THR1010 and ARG880 participate in hydrogen bonding, and PHE1009 and PHE914 form Pi–Pi interactions. The H bond interactions were formed between catechin and THR 1010 and ARG880. LEU1014 and LEU873 were involved in the Pi–sigma interactions, and PHE1009 and PHE 914 were involved in the Pi–Pi T-shaped interactions. In summary, the phenolic hydroxyl and carbonyl groups of the components in MOL-FP form hydrogen bonds with THR1010, ARG880, and SER876. Pi–Pi Stacked interactions were formed between aromatic ring and PHE914.

### 2.5. Metabolomic Analysis

The study of serum metabolite variation aims at identifying early and differential metabolic markers of diseases and evaluating the effects of treatment with drugs/functional foods on diseases. These metabolites associated with diseases and therapies become targets for the prevention and treatment of chronic diseases. To systematically illustrate mechanisms underlying the amelioration effect of MOL-FP on hyperuricemia, UPLC-MS-based metabolomics was applied. Principal component analysis (PCA) is a representative of unsupervised algorithms, which can reflect the spatial distribution of the original data of the samples without any artificial factor grouping. The closer the spatial distribution of the samples to each other, the closer the composition of the metabolite components within the samples [22,23,24]. In the current research, PCA was carried out to discern the differences of metabolite fingerprints in serum among NG, MG, and DG. The separation trend of NG, MG, and DG based on metabolic data was observed in both negative and positive ion modes (Figure 5A,B), indicating that the group difference was more remarkable than the individual difference and endogenous metabolites in serum were significantly different.

### 2.6. Identification of Potential Endogenous Biomarkers

OPLS-DA is a supervised discriminant analysis statistical method that uses partial least squares regression to model the relationship between metabolite expression and sample categories to achieve the prediction of sample categories. OPLS-DA and independent samples T-tests were performed to indicate the difference among NG, MG, and DG to find potential biomarkers of the effects of HUA and the therapeutic effects of MOL-FP. DG and MG showed a clear separation into two classes in both positive and negative modes in OPLS-DA score plots (Figure 5C,D), revealing that MOL-FP administration affected metabolic pathways in hyperuricemia mice. Model validation with the number of permutations equaling 200 generated intercepts of R2 = 0.743 and Q2 = −0.382 in positive mode and R2 = 0.601 and Q2 = −0.565 in negative mode (Figure 5E,F), suggesting that the OPLS-DA models were reliable and not over-fitting.

S-plots visualizing the relationship between covariance and correlation from the OPLS-DA model and variable importance in the projection (VIP) value were employed to identify the features contributing to group separation. The X-axis indicates the contribution of the metabolite, while the Y-axis indicates the correlation of the sample within the same sample group. Therefore, the variables contributed highly to the differentiation to be located at both ends of the S-shaped curve. In the present study, these variables with a VIP value > 1.0 were tested by independent samples *t*-test. The serum metabolites (VIP value > 1.0; *p* < 0.05) were considered as potential biomarkers that correlated with the therapeutic effects of MOL-FP on hyperuricemia mice (Figure 5G,H). The red dot means VIP > 1.0. The possible molecular formula of the potential biomarker was calculated using high-accuracy quasi-molecular ions within a mass error of 10 ppm and fractional isotope abundance was detected by UPLC-MS. These potential biomarkers were presumed based on accurate elemental composition and MS/MS behavior. The structure information was obtained by searching freely accessible databases mentioned above. A total of 38 potential endogenous biomarkers were identified as summarized in Appendix A. Figure 6 shows the change in their quantities in the form of a heat map.

### 2.7. Analysis of Metabolic Pathway of Potential Biomarkers

Pathway analyses were used to evaluate the role of metabolites in biological reactions based on the position of metabolites in related pathways, and to identify the involved metabolomic pathways. To evaluate the therapeutic effect and interpret the mechanisms of MOL-FP on hyperuricemia in mice, we investigated the response of differential metabolites perturbed by the model establishment after being treated with MOL-FP. The differential metabolites were imported to MetaboAnalyst 5.0 for the metabolic pathway analysis. These results showed that arginine biosynthesis; glycerophospholipid metabolism; D-glutamine and D-glutamate metabolism; biosynthesis of unsaturated fatty acids; purine metabolism; tyrosine metabolism; phenylalanine, tyrosine, and tryptophan biosynthesis; taurine and hypotaurine metabolism, etc. in PO and HX-induced hyperuricemia mice were affected by MOL-FP treatment (Figure 7A,B).

## 3. Discussion

Hyperuricemia is a metabolic disease caused by disorders of purine metabolism. At present, the main clinical drugs are allopurinol and febuxostat, which inhibit uric acid production, and benzbromarone, which promotes uric acid excretion. Although these drugs have significant effects in the treatment of uric acid reduction, serious liver and kidney damage and toxic side effects have limited their clinical application [25]. Therefore, it is valuable to develop new drugs or functional foods with few toxic side effects. Flavonoids and phenolics have been proposed to have great potential in the treatment of hyperuricemia [18]. 

In our previous study, we obtained MOL-FP by extraction and purification. The chemical composition was identified as 3-*p*-coumaroylquinic acid, 5-*p*-coumaroylquinic acid, astragalin, catechin, chlorogenic acid, isoquercetin, isovitexin, marumoside B, niazirin, and vicenin-2 by UPLC-MS (Appendix A and Appendix A) [26]. In the present study, we demonstrated that MOL-FP had a significant effect on reducing serum UA by inhibiting XO activity and regulating renal urate transporter. The inhibition of xanthine oxidase activity is an important treatment to reduce uric acid production. MOL-FP treatment significantly decreased serum and hepatic XO activity of hyperuricemia mice. Molecular docking virtual screening results showed that 3-*p*-coumaroylquinic acid, 5-*p*-coumaroylquinic acid, and catechin had an excellent affinity to the xanthine oxidase, indicating that these components are potential inhibitors of xanthine oxidase. In addition, astragalin, chlorogenic acid, isoquercetin, and isovitexin also have good affinity to the xanthine oxidase (Appendix A). Down-regulation of uric acid reabsorption transporters and up-regulation of uric acid secretion transporters are vital for the promotion of uric acid excretion. The expression of urate reabsorption proteins (URAT1 and GLUT9) was reduced and secretion proteins (ABCG2, OAT1, and OAT3) were increased in the kidneys of hyperuricemia mice after MOL-FP treatment, indicating that MOL-FP had an inhibitory effect on UA reabsorption and facilitated excretion in hyperuricemia mice. In addition, MOL-FP also showed significant hepato-renal protective effects. The effect of phenolics and peptides from *Moringa oleifera* Lam. leaf hydrolysate on reducing UA also has been reported [19], which is consistent with our results.

Metabolomics has been widely applied as a novel and holistic diagnostic tool in clinical and biomedical research [27]. Due to the continuous development of advanced analytical techniques and bioinformatics, metabolomics is commonly implemented to understand the pathophysiological processes involved in disease progression as well as to find new diagnostic or prognostic biomarkers for diseases of various organisms [28]. In the present study, 38 potential endogenous biomarkers were identified based on UPLC-ESI-Q-Exactive/MS. These 38 metabolites are mainly related to purine metabolism, amino acids metabolism, and lipid metabolism.

Purine metabolism disorder is one of the main reasons for hyperuricemia [29]. Inosinic acid, which belongs to the class of organic compounds known as purine ribonucleoside monophosphates, is metabolized to uric acid catalyzed by a multitude of enzymes [30]. In our studies, levels of hypoxanthine, inosinic acid, and UA in the Model group are significantly higher than those in the normal group, indicating that purine metabolism is disrupted. The increase in hypoxanthine might be related to the long-term administration of PO and HX. In the MOL-FP group, the inosinic acid and UA levels are closer to those of the Normal group, and the increase of hypoxanthine demonstrates that the synthesis of UA is reduced, indicating that MOL-FP could regulate the disorder of purine metabolism. 

Amino acids (AAs) are fundamental components of living organisms and have structural and active dynamic roles in the physiology of tissues and cells [31]. AAs are involved in various biochemical processes, including the biosynthesis of uric acid. In this study, we found that the levels of Taurine, L-cysteine, N-Acetylhistamine, L-Lysine, Tyrosine, L-Glutamine, L-Glutamic acid, Glycine, and L-Arginine in the serum of the HUA mice were significantly altered than those in normal mice, suggesting the metabolism of amino acids was disrupted in the mice with HUA. After treatment with MOL-FP, levels of the above metabolites were close to those of the normal group. Taurine, the product of the metabolism of the sulfur-containing amino acids L-cystine and L-cysteine, has many diverse biological functions, including serving as a neurotransmitter in the brain, a stabilizer of cell membranes, and a facilitator in the transport of ions, such as sodium, potassium, calcium, and magnesium [32,33]. It is worth noting that taurine might play a role in the regulation of renal uric acid excretion [32]. The metabolites, such as tyrosine, metamachine, and normetanephrine, are related to tyrosine metabolism. Tyrosine can be metabolized into neurotransmitters such as L-DOPA, dopamine, adrenaline, or noradrenaline. Several reports have indicated that L-dopa may cause hyperuricemia and gout by inhibition of uric acid excretory transport [34]. Glutamine is an essential nutrient to maintain renal function and is the main donor of NH3 in the kidney, which plays an important role in the regulation of renal acid–base balance, and its decreased serum level may indicate abnormal renal function [35]. Glycine is the synthetic precursor of urate, and a large amount of glycine is used in the synthesis of urate [36]. In addition, glycine inhibits hypoxia-inducible factor-1α by inhibiting the up-regulation of nuclear factor-κb p65, thereby inhibiting the inflammatory response. The decrease in plasma glycine level can lead to the weakening of the body’s anti-inflammatory ability [37]. Combined supplementation with glycine and tryptophan significantly decreased serum uric acid levels, suggesting that the metabolism of glycine and tryptophan was related to hyperuricemia [38]. L-Arginine can catalyze the circulation of ornithine, promote the formation of urea, and make ammonia in the human body become non-toxic urea [39]. It is a substrate for the endogenous synthesis of nitric oxide, which is generated under the catalysis of synthetase and exerts physiological effects. This biochemical process is called the nitric oxide pathway. Nitric oxide as an intercellular messenger and neurotransmitter plays an important role in the cardiovascular system, central nervous system, and peripheral transmission [40]. Hyperuricemia rats have a decrease in serum nitric oxide which is reversed by lowering uric acid levels [41]. L-Arginine confers protection against hepatorenal damage via the suppression of uric acid generation and blockade of glutathione dysregulation [42]. This is evidence that arginine metabolism is closely associated with hyperuricemia. From the biomarkers found in metabolomics, Taurine, L-cysteine, L-Glutamine, and L-Glutamic acid decreased in the model group compared to the normal group and could be up-regulated after MOL-FP treatments. Glycine, N-Acetylhistamine, L-Lysine, Tyrosine, Metanephrine, and Normetanephrine increased in the model group and decreased after MOL-FP treatment. The results demonstrated that Amino acids metabolism, such as taurine and hypotaurine metabolism, tyrosine metabolism, D-Glutamine and D-glutamate metabolism, glyoxylate and dicarboxylate metabolism, and arginine biosynthesis, were seriously disturbed in hyperuricemia mice, and MOL-FP could alleviate hyperuricemia through regulating the disorder of Amino acids metabolism in vivo.

Studies have shown that Hyperuricemia is associated with lipid metabolism disorders [43]. In this study, we found that MOL-FP could regulate lipid metabolism disorders in hyperuricemia mice. Phospholipids were divided into sphingolipids and glycerophospholipids. According to the different functional groups of the phosphate head, GP could be divided into PC, PE, PG, PS, PA, and PI, these could be hydrolyzed to free fatty acid (FFA) and lysophospholipid [44]. CE, PE, LysoPA, LysoPC, and LysoPE are mainly involved in the metabolism of glycerophospholipids, and their levels were significantly down-regulated in HUA mice and up-regulated in the MOL-FP group. Glycerophospholipids are biologically active lipids, often referred to as polar lipids, that play an essential role in maintaining the integrity and function of membranes. As second messengers and precursors of bioactive lipids, they also play an important role in cell signaling processes and may be involved in the pathogenesis of inflammatory and cardiometabolic diseases [45]. Lysophosphatidylcholine is found in small amounts in most tissues. It is formed by the hydrolysis of phosphatidylcholine by the enzyme phospholipase A2, as part of the de-acylation/re-acylation cycle that controls its overall molecular species composition. In blood or plasma, LPCs are bound mainly to albumin and a lesser extent to lipoproteins. Inflammation, cell damage, and other pathophysiological conditions can profoundly alter the ratio of free to albumin-bound LPC through increased production of LPC or decreased plasma levels of albumin [46]. Unsaturated fatty acids (UFAs) have functions in immune regulation, as shown by animal research and human clinical medicine [47]. The levels of eicosapentaenoic acid, docosapentaenoic acid (22n-6), linolenelaidic acid, oleic acid, eicosadienoic acid, docosahexaenoic acid, docosatrienoic acid, erucic acid, and arachidonic acid in the hyperuricemia group were significantly decreased compared to the normal group suggesting that Biosynthesis of unsaturated fatty acids was disturbed. In conclusion, lipid metabolites, such as glycerophospholipid metabolism or biosynthesis of unsaturated fatty acids, were basically in the state of metabolic inhibition. In addition, we found that MOL-FP could alleviate the disorder of lipid metabolism in hyperuricemia mice in varying degrees of reversal of the average level of potential biomarkers, and there was no significant difference in phospholipid metabolism to the normal group. These data showed that MOL-FP had a unique effect on hyperuricemia caused by lipid metabolism disorder.

## 4. Materials and Methods

### 4.1. Reagents

Potassium oxonate, Hypoxanthine, Xanthine, and XO were purchased from Shanghai yuan ye Bio-Technology Co., Ltd. (Shanghai, China). Assay kits for the determination of urea nitrogen (BUN), creatinine (CRE), Aspartate aminotransferase (AST), and Alanine aminotransferase (ALT) were obtained from Nanjing Jiancheng Bioengineering Inc., (Nanjing, China) Biotechnology. 0.9% NaCl injection was obtained from Anhui Shuanghe Pharmaceutical Co., Ltd. (Anhui, China). The uric acid Kit was purchased from Biosino Biotechnology and Science inc (Beijing, China). HPLC-grade methanol and acetonitrile were purchased from Merck (Beijing, China). Ultrapure water was prepared by A.S. Watson TM Limited (Beijing, China).

### 4.2. Plant Materials and Preparation of MOL-FP

The Sun-dried leaves of *Moringa oleifera* Lam. were purchased from Anhui Jiulixiang Pharmaceutical Co., Ltd. (Anhui, China) and authenticated by Professor Chunsheng Liu at the Beijing University of Chinese Medicine, Beijing, China. A voucher specimen (No. 20190910) was deposited in the herbarium of the Beijing University of Chinese Medicine. MOL was ground into a fine powder, sieved with a 60-mesh sieve, and stored at −20 °C until use. The 1000 g of dried raw powder was soaked twice with 20 times the volume of 60% ethanol (1:10, *w*/*v*) at 80 °C for 1.5 h each. Then, the extracts were combined and filtered. The combined filtrates were concentrated by rotary evaporation at 40 °C under vacuum to eliminate ethanol. Then, water was added to adjust the volume to 5000 mL (concentration of raw material was 0.2 g/mL), vacuum filtered, centrifuged at 4000 rpm for 20 min, and the filtrate was loaded onto an NKA-9 macroporous resin column with a bed volume (BV) of 5000 mL and successively eluted with 2 BV water to elute the impurities and 7 BV 70% EtOH to elute the fractions of flavone solution. The eluting velocity was 6 BV/h, and the 70% ethanol fraction was evaporated in vacuo and then dried under vacuum at 40 °C. Finally, the resultant powder was stored in a desiccator until use. The extract yield of the flavone fraction was 15.6% (*w*/*w*). In our previous study, the main compositions of MOL-FP was identified as 3-*p*-coumaroylquinic acid, 5-*p*-coumaroylquinic acid, astragalin, catechin, chlorogenic acid, isoquercetin, isovitexin, marumoside B, niazirin, and vicenin-2 by UPLC-MS [26] (Appendix A).

### 4.3. Animals and Drug Administration

Male KM mice (20 ± 2 g) were purchased from Beijing Vital River Laboratory Animal Technology Co., Ltd. (Beijing, China; certification no: SCXK (Jing) 2016-0011) and housed under a 12 h light/dark cycle for 5 days to adapt to the environment with controlled temperature (24 ± 1 °C) and humidity (45 ± 5%). All mice were given free access to food and water. Our protocols have followed the guidelines established by the

BUCM Animal Care Committee. All doses were expressed as milligrams per kilogram (mg/kg) body weight of the respective drugs. The dose administered was based on body weight measured before administration. Allopurinol (AP), potassium oxonate (PO), hypoxanthine (HX), and MOL-FP were suspended in 0.9% physiological saline solution.

The mice were randomly divided into 6 groups (10 mice in each group): control group (Con), HUA model group (Model), positive drug (allopurinol) group (AP), low-dose MOL-FP group (600 mg/kg), medium-dose MOL-FP group (1200 mg/kg), and high-dose MOL-FP group (2400 mg/kg). The hyperuricemia mice were generated via the intraperitoneal (i.p.) administration of 300 mg/kg PO and oral gavage (p.o.) of 300 mg/kg HX for all groups except the Control group. The Control group was treated with the same volume of 0.9% physiological saline solution, once a day for 7 consecutive days. On the 8th day, one hour after modeling, mice in AP group were orally given allopurinol (10 mg/kg) and mice in drug groups orally received MOL-FP (600, 1200 and 2400 mg/kg) once daily for 1 week, respectively. The body weight was measured every 7 days.

Based on the above studies, the effect of MOL-FP on the endogenous metabolism of hyperuricemia mice was researched. Thirty mice were employed and divided into three groups (10 mice per group), a normal control group (NG), a model group (MG), and MOL-FP treated group (DG). The dosages of MOL-FP were 1200 mg/kg, and drug administration was performed by gavage once a day for 30 days.

### 4.4. Sample Collection and Preparation

The whole blood sample was collected from the orbital venous plexus (1.5 mL blood sample for each mouse), allowed to clot for approximately 1 h at room temperature, and centrifuged (3500 rpm, 10 min). The supernatants were transferred to tubes and stored at −80 °C until analysis. The livers and kidneys were excised on an ice plate and washed in phosphate-buffered saline (PBS), promptly frozen with liquid nitrogen, and stored at −80 °C until use. The part of the liver and renal tissues of mice fixed in 10% formalin. The tissues were dehydrated in graded ethanol solutions and embedded in paraffin. Tissue sections were cut and stained with hematoxylin-eosin (HE) for conventional morphological evaluation, respectively. Serum uric acid (SUA), blood urea nitrogen (BUN), creatinine (CRE), and serum XO activity were measured using kits. Renal ABCG2, GLUT9, and URAT1 RNA expressions were detected by RT-qPCR.

In serum metabolomics studies, 100 μL of serum samples were thawed at 4 °C before analysis. Serum proteins were precipitated with 300 μL of methanol and removed by centrifugation (12,000× *g* at 4 °C, 15 min). The supernatants were dried under a nitrogen stream at room temperature and redissolved in 100 μL of methanol–water (80:20, *v*/*v*). Each reconstituted sample was centrifuged (12,000× *g* at 4 °C, 15 min) to transfer the supernatant into a new vial. Additionally, 10 μL from each serum was pooled and processed by the same method to gain a quality control sample (QC).

### 4.5. Molecular Docking

The molecular docking screening was conducted using AutoDock Vina1.2.0 [48,49]. XO crystal structure (PDB ID, 3nvw) was set as the receptor, in which Guanine was selected as the active site (center_x = 37.733, center_y = 20.262, center_z = 18.48; size_x = 20, size_y = 20, size_z = 20) and then removed. Their 3D structures are illustrated and energetically minimized using ChemDraw 3D. Docking was executed using the default parameters to obtain poses that were refined using the annealing molecular dynamics. Poses with the highest affinity were selected for further analysis. The better affinity with XO is considered if the binding energy of components in MOL-TF is lower than that of Guanine. Pymol and Discovery Studio were applied for analysis and visualization. 

### 4.6. UPLC-MS Instrument Conditions

UPLC analysis system was equipped with an ACQUITY UPLC HSS T3 column (2.1 × 100 mm, 1.8 μm, Waters, Ireland) fitted with an ACQUITY HSS T3 1.8 mum VanGuard™ Pre-Column (2.1 × 5 mm). The column temperature was 30 °C, and the sample injection volume was 4 μL. The mobile phase consisted of 0.1% (*v*/*v*) formic acid (A) and acetonitrile (B). The gradient elution was performed as follows: 2% B at 0–1 min, 2–5% B at 1–2 min, 5–25% B at 2–5 min, 25–75% B at 5–8 min, 75–95% B at 8–12 min, and 95% B isocratic for another 2 min. The flow rate was 0.4 mL/min. 

Mass spectrometry was performed on a Thermo-Fisher Scientific Q-Exactive Orbitrap mass spectrometer equipped with a HESI source. The ion source parameters were set as follows: capillary and aux gas heater temperature, 300 and 350 °C, respectively; spray voltage, 3.5 kV in positive mode and 3.2 kV in negative mode; sheath gas (N2) was 35 arbitrary units (arb) and auxiliary gas (N2) was 15 arb (N2, 99.9% purity). The normalized collision energy (NCE) was ±35 V. The scanning range was *m*/*z* 70–1050 under full-MS/dd-MS2.

### 4.7. Data Analysis and Identification of Potential Biomarkers

Raw data were processed by Progenesis QI software v2.0 (Nonlinear Dynamics, Newcastle upon Tyne, UK) for peak detection, peak alignment, normalization, and other operations. The pretreated data were, respectively, uploaded into SIMCA-P 14.1 software (Umetrics AB, Umea, Sweden) for multivariate data analysis. Principal component analysis (PCA) revealed the distribution of metabolites in mouse serum samples. Orthogonal partial least-squares discriminant analysis (OPLS-DA) models were established to distinguish sample differences and mine differential metabolites in massive data. Permutation tests were used to verify the validity of the OPLS-DA model. The variable importance in the projection (VIP) value of each variable in the OPLS-DA model was calculated to indicate its contribution to the classification. Metabolites with VIP greater than 1, *p* values of *t*-test (*p*) less than 0.05, and a fold change (FC) of greater than 2.0 or FC less than 0.5 were selected as differential metabolites [50].

The chemical information of differential metabolites was searched through the human metabolome database (HMDB; http://www.hmdb.ca/; accessed on 10 September 2022.) and METLIN (http://metlin.Scripps.edu; accessed on 20 September 2022.). For exploring how the major metabolic pathways related to the differential metabolites were affected, the metabolic pathway was analyzed by MetaboAnalyst 5.0 platform (http://www.metaboanalyst.ca; accessed on 15 October 2022.). The metabolic pathways, pathway impact > 0.1 and *p* < 0.05, are usually considered the most relevant metabolic pathways. A heatmap plot was constructed using the R package program.

### 4.8. Statistical Analyses

Statistical analysis was performed using GraphPad Prism 9 (GraphPad Software, San Diego, CA, USA) was performed by one-way analysis of variance (ANOVA) to determine the levels of statistical significance. Significant differences emerging from the above tests are indicated in the figures by # *p* < 0.05, ## *p* < 0.01, ### *p* < 0.001 (vs. Model group) and * *p* < 0.05, ** *p* < 0.01, *** *p* < 0.001 (vs. Control group).

## 5. Conclusions

In summary, we obtained MOL-FP by extraction and purification and demonstrated that MOL-FP had a significant effect on reducing serum UA by inhibiting serum and liver XO activity, regulating renal ABCG2, URAT1, and GLUT9 expressions. In silico molecular docking analysis indicated that 3-p-coumaroylquinic acid, 5-p-coumaroylquinic acid, and catechin are potential inhibitors of xanthine oxidase. By profiling metabolites in the serum of hyperuricemia mice using UPLC-ESI-Q-Exactive/MS-based metabolomics analysis, we highlighted that MOL-FP effectively ameliorated hyperuricemia by regulating Purine metabolism, amino acid metabolism, and lipid metabolism. This study illustrates that MOL-FP is very promising as a functional food ingredient for the prevention and treatment of hyperuricemia.

## Figures and Tables

**Figure 1 molecules-27-08237-f001:**
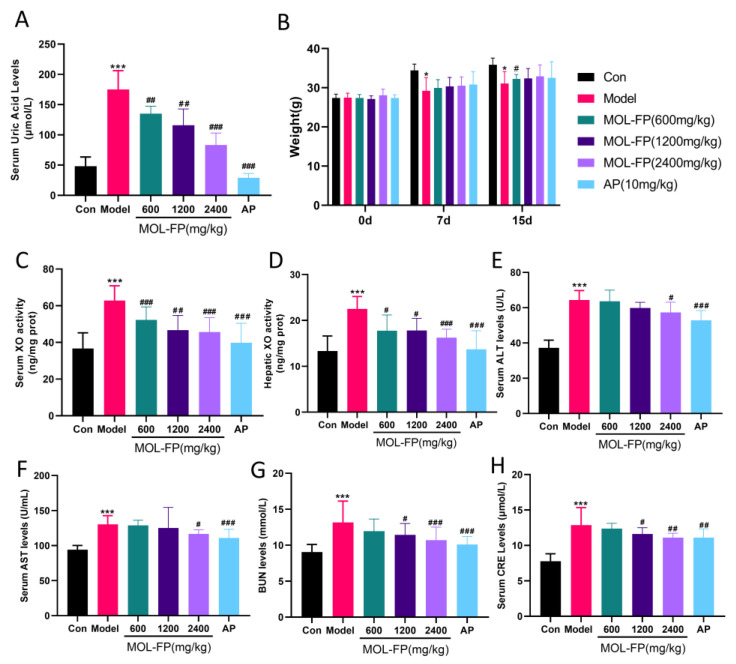
Effect of MOL-FP on (**A**) body weight, (**B**) serum UA level, (**C**) serum XO activity, (**D**) hepatic XO activity, (**E**) serum ALT level. (**F**) serum ALT level. (**G**) serum BUN level. (**H**) serum CRE level. All values are mean ± standard deviation (*n* = 10). * *p* < 0.05 and *** *p* < 0.001 compared with the Control group. # *p* < 0.05, ## *p* < 0.01 and ### *p* < 0.001 compared with the Model group.

**Figure 2 molecules-27-08237-f002:**
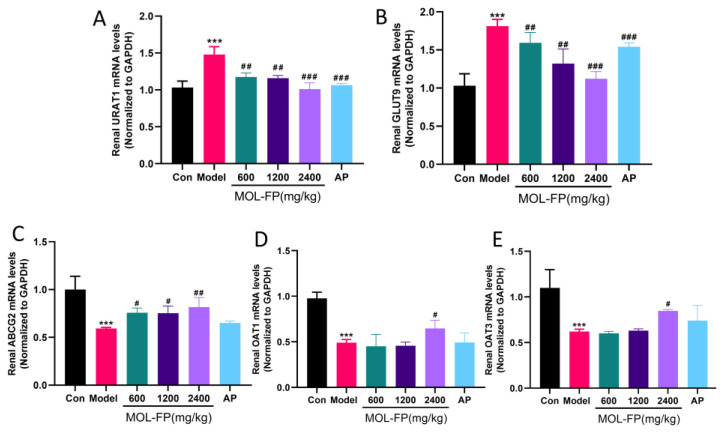
Effect of MOL-FP on (**A**) URAT1, (**B**) GLUT9, (**C**) ABCG2, (**D**) OAT1, and (**E**) OAT3 of the hyperuricemic mice. All values are mean ± standard deviation (*n* = 10). *** *p* < 0.001 compared with the Control group. # *p* < 0.05, ## *p* < 0.01 and ### *p* < 0.001 compared with the Model group.

**Figure 3 molecules-27-08237-f003:**
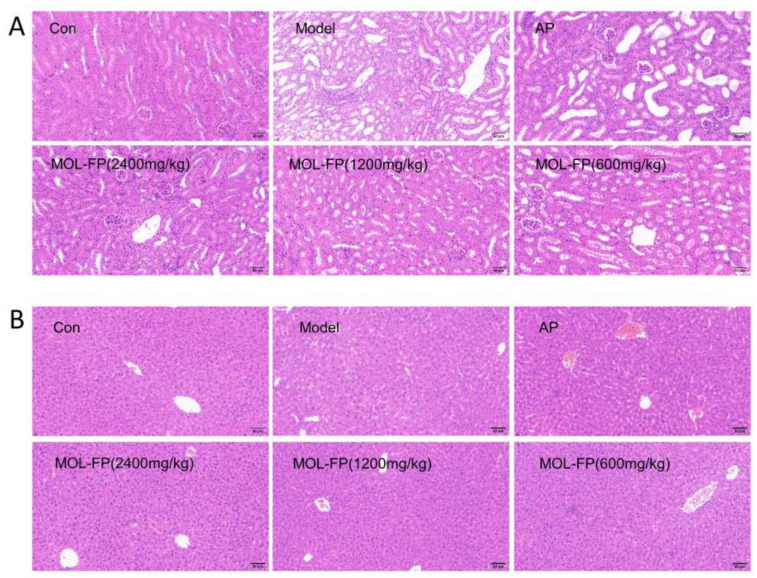
Representative H&E stained sections of mouse kidney (**A**) and liver (**B**). Scale bar = 60 μm, original magnification 400×.

**Figure 4 molecules-27-08237-f004:**
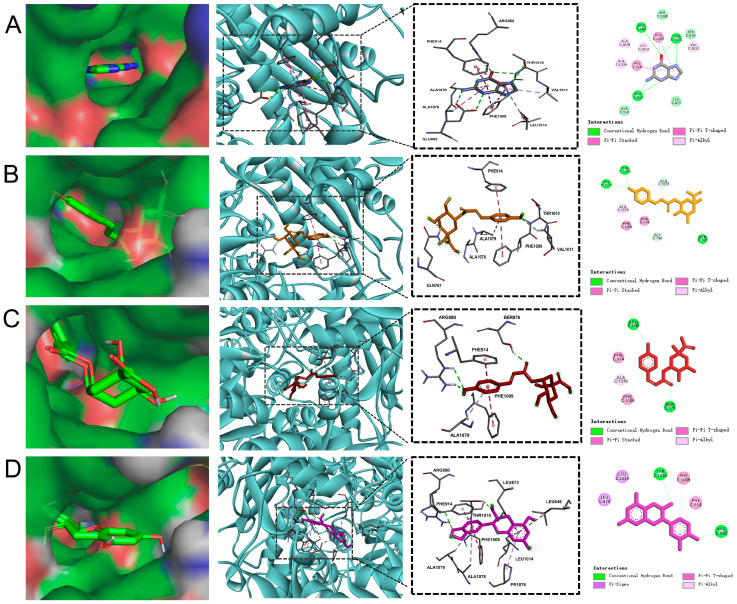
The result of molecular docking: (**A**) Guanine; (**B**) 5-*p*-Coumaroylquinic acid; (**C**) 3-*p*-Coumaroylquinic acid; (**D**) Catechin.

**Figure 5 molecules-27-08237-f005:**
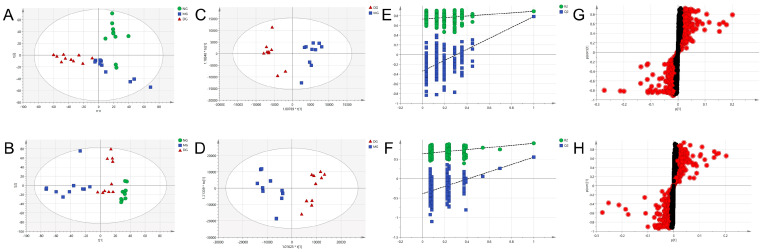
Metabolomic analyses of serum samples. (**A**,**B**) PCA score plots of the three groups from the normal, the model, and the drug-treated group in positive mode and negative mode. (**C**,**D**) OPLS-DA score plots of the model and the drug-treated group based on the serum metabolic profiles in positive mode and negative mode. (**E**,**F**) permutation test of serum samples from the model and the drug-treated group in positive mode and negative mode. (**G**,**H**) S-plot model group and MOL-FP treated from the OPLS-DA model in positive mode and negative mode. The red dot means VIP > 1.0.

**Figure 6 molecules-27-08237-f006:**
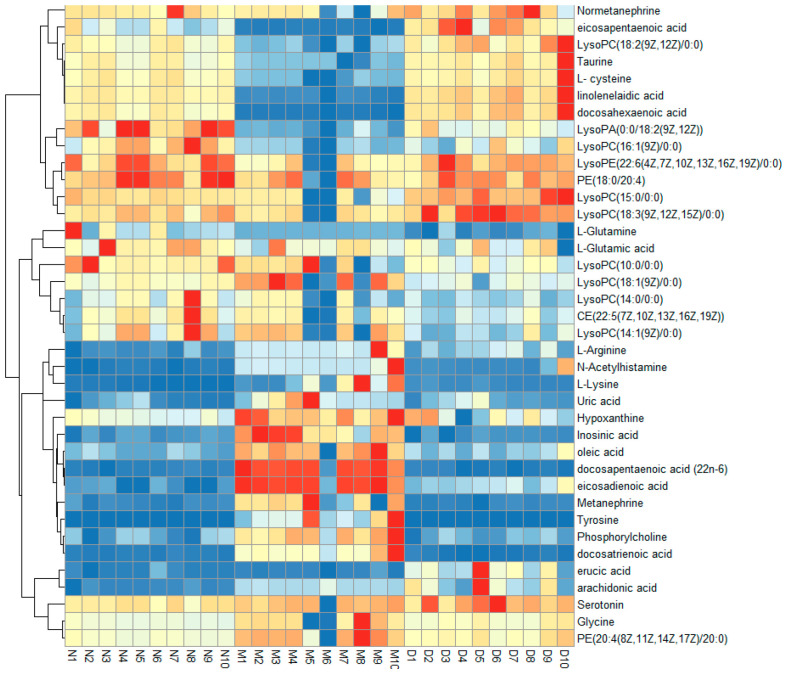
Hierarchical clustering heat map of the 38 differential metabolites, with the degree of change marked in red (up-regulation) and blue (down-regulation).

**Figure 7 molecules-27-08237-f007:**
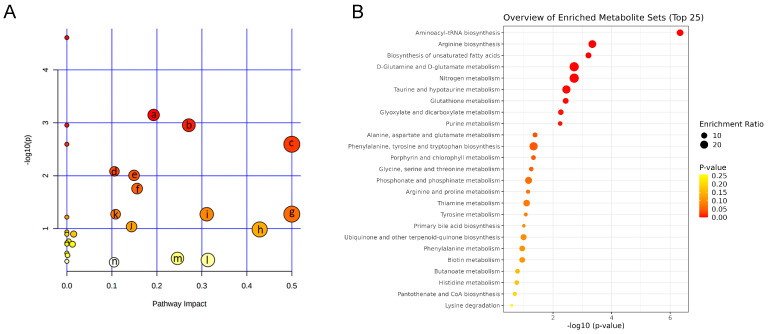
(**A**) Metabolic pathway analysis of 38 differential metabolites. (a) Arginine biosynthesis; (b) Glycerophospholipid metabolism; (c) D-Glutamine and D-glutamate metabolism; (d) Biosynthesis of unsaturated fatty acids; (e) Purine metabolism; (f) Tyrosine metabolism; (g) Phenylalanine, tyrosine and tryptophan biosynthesis; (h) Taurine and hypotaurine metabolism; (i) Alanine, aspartate and glutamate metabolism (j) Arginine and proline metabolism; (k) Glutathione metabolism; (l) Arachidonic acid metabolism; (m) Glycine, serine and threonine metabolism; (n) Tryptophan metabolism (**B**) Analysis of enrichment pathway of altered metabolites.

## Data Availability

Data are contained within the article.

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
