# Peer review of "The Therapeutic Effect and the Potential Mechanism of Flavonoids and Phenolics of Moringa oleifera Lam. Leaves against Hyperuricemia Mice"

_molecules, 2022, doi:10.3390/molecules27238237_

Round 1

Reviewer 1 Report

In the study, the authors reported the MOL-FP extract that showed a significant effect on reducing serum UA by inhibiting serum and liver XO activity, regulating renal ABCG2, URAT1and GLUT9 expressions. The manuscript is well prepared and the results are supportive to their claims. The study is comprehensive with different kind of experimental results to support the effects of the MOL-FP extract on the XO activity. The study is generally interesting and could be suitable for publication. There are some issues to be addressed by the authors.

1.  Too many symbols such as NC, M, AP…, did not explain well and that makes the manuscript difficult to read.

2. The study is focused on the investigation of the inhibition of XO. The protein level of XO did not evaluate with Western blotting assay. The MOL-FP extract should be multi-target. The protein level of XO should be checked.

3. The names of authors are not consistent for manuscript and Supporting Information.

4. Some typo errors found.

5. Animal ethics approval for the study needs to be obtained.

Author Response

Point 1: Too many symbols such as NC, M, AP, did not explain well and that makes the manuscript difficult to read.

Response 1: Thanks for your kind concern. We are sorry that the manuscript is difficult to read because of symbols did not explain well. We have added a description of symbols to the revised manuscript. The symbols of Figure1, Figure2, and Figure3 have also been modified.

Point 2: The study is focused on the investigation of the inhibition of XO. The protein level of XO did not evaluate with Western blotting assay. The MOL-FP extract should be multi-target. The protein level of XO should be checked.

Response 2: Thank you for your valuable comments. In this study, The serum and hepatic XO activities of hyperuricemic mice significantly decreased after MOL-FP treatment, but we have no evidence that MOL-FP can reduce the expression of the protein level of XO. Your excellent suggestions provide direction for our further research. In future studies, we will focus on the effect of MOL-FP on XO protein expression.

Point 3: The names of authors are not consistent for manuscript and Supporting Information.

Response 3: Thanks for your kind reminder. The names of authors in Supporting Information have been corrected and are consistent with the manuscript.

Point 4: Some typo errors found.

Response 4: Thanks for your kind reminder. We have carefully checked and corrected these errors.

Point 5: Animal ethics approval for the study needs to be obtained.

Response 5: Thanks for your kind reminder. In fact, this study was approved by The Experimental Animal Ethics Committee of the Academic Committee of Beijing University of Chinese Medicine (project identification code: BUCM-4-2022013101-1111). We have added the Animal ethics approval in the part of “Institutional Review Board Statement”

Reviewer 2 Report

Moringa oleifera Lam. Leaves extracts are well known for their therapeutic activities in several diseases, in this study authors specifically evaluate the anti-hyperuricemia effect and clarify the possible mechanisms of flavonoids and phenolics of MOL (MOL-FP) in mice. The study is nicely performed and written however I would like to give some suggestions to improve the quality of work.

Below are some points of suggestion, kindly address them accordingly:

1.      Kindly provide the global morbidity and mortality related to hyperuricemia in the introduction section.

2.      There are several studies highlighting the therapeutic potential (anti-hyperuricemia) of Moringa oleifera fractions, however, I noticed that in the introduction section of this manuscript there is no information given related to this. Therefore, kindly provide brief information about the previous reports and what is novel in this current study with citations.

3.      The author did not mention about ethical clearance for this study, kindly provide the ethical approval number.

4.      Kindly provide the LD50 dose of Moringa oleifera fractions with proper citation, and explain why these three doses (600, 1200, and 2400 mg/kg) were chosen for this study.

5.      In the molecular docking study kindly provide the active site residues information or grid box dimensions with XYZ coordinates of the grid.

6.      Kindly improve the quality of images, as in molecular docking images the name and id of amino acid residues are not clearly visible

7.      Kindly provide the binding energy of the reference standard and compare the interacting residues of native ligand, reference standard, and studied compounds.

8.      Some information like year and volume is not available in reference number 17. 

Author Response

Point 1: Kindly provide the global morbidity and mortality related to hyperuricemia in the introduction section.

Response 1: Thank you for your comments. The global morbidity and mortality related to hyperuricemia have been supplemented in the revised manuscript(1. Introduction).

Point 2: There are several studies highlighting the therapeutic potential (anti-hyperuricemia) of Moringa oleifera fractions, however, I noticed that in the introduction section of this manuscript there is no information given related to this. Therefore, kindly provide brief information about the previous reports and what is novel in this current study with citations.

Response 2: Thank you for your comments. The information about the previous reports and innovations of the current study have been supplemented in the revised manuscript(1. Introduction).

Point 3: The author did not mention about ethical clearance for this study, kindly provide the ethical approval number.

Response 3: Thanks for your kind reminder. In fact, this study was approved by The Experimental Animal Ethics Committee of the Academic Committee of Beijing University of Chinese Medicine (project identification code: BUCM-4-2022013101-1111). We have added the Animal ethics approval in the part of “Institutional Review Board Statement” .

Point 4: Kindly provide the LD50 dose of Moringa oleifera fractions with proper citation, and explain why these three doses (600, 1200, and 2400 mg/kg) were chosen for this study.

Response 4: Thank you for your kind reminder. Most flavonoids and polyphenols are low toxic (LD50 > 5 g/kg)[1,2]. Before the formal experiment, we carried out two pre-experiments: (1). Mice were orally given MOL-FP (5g/kg) test for 14 days, and did not provoke acute toxicity. (2). The effective dose of MOL-FP was 600 mg/kg. On this basis, the medium dose was set to 1200mg/kg and the high dose to 2400mg/kg.

Point 5: In the molecular docking study kindly provide the active site residues information or grid box dimensions with XYZ coordinates of the grid.

Response 5: Thanks for your kind reminder. The information about grid box dimensions has been supplemented in the revised manuscript(4.5. Molecular Docking).

Point 6: Kindly improve the quality of images, as in molecular docking images the name and id of amino acid residues are not clearly visible

Response 6: Thanks for your kind reminder. The molecular docking images have been resubmitted, hoping to meet the quality request of Molecules(Figure 4).

Point 7: Kindly provide the binding energy of the reference standard and compare the interacting residues of native ligand, reference standard, and studied compounds.

Response 7: Thank you for your valuable comments. It has been supplemented the binding energy of the reference standard and compare the interacting residues of native ligand, reference standard, and studied compounds in the revised manuscript.

Point 8: Some information like year and volume is not available in reference number 17.

Response 8: Thanks for your kind reminder. The format of the references has been modified in the revised manuscript, reference number 17 was revised to number 26, which has not been published at present, its website address has been linked in the end. Some information like year has been supplemented.

References

  1. Karbab, A.; Mokhnache, K.; Ouhida, S.; Charef, N.; Djabi, F.; Arrar, L.; Mubarak, M.S. Anti-Inflammatory, Analgesic Activity, and Toxicity of Pituranthos Scoparius Stem Extract: An Ethnopharmacological Study in Rat and Mouse Models. J Ethnopharmacol 2020, 258, 112936, doi:10.1016/j.jep.2020.112936.
  2. Fujii, H.; Sun, B.; Nishioka, H.; Hirose, A.; Aruoma, O.I. Evaluation of the Safety and Toxicity of the Oligomerized Polyphenol Oligonol. Food Chem Toxicol 2007, 45, 378–387, doi:10.1016/j.fct.2006.08.026.